# Investigation of Vegetable Oils and Their Derivatives for the Synthesis of Extreme Pressure Additives

**DOI:** 10.3390/ma16196570

**Published:** 2023-10-06

**Authors:** Gábor Zoltán Nagy, Roland Nagy

**Affiliations:** Department of MOL Hydrocarbon and Coal Processing, Faculty of Engineering, University of Pannonia, 8200 Veszprém, Hungary

**Keywords:** EP additives, vegetable oil, vegetable oil derivates, synthesis

## Abstract

The harmful effects of wear can be reduced through proper lubrication of the frictional parts. When exposed to excessive loads, the lubricant film is displaced from the surfaces, and even the adhesive lubricant layer may rupture. Additives known as Extreme Pressure (EP) are frequently incorporated into lubricants to minimise wear and avert seizures under high temperature and pressure. Mechanistically, these additives generate a film on the surface through chemisorption. These additives are extensively applied in various lubricants, with the largest quantities being employed in metalworking fluids and lubricating greases. Sulfurized vegetable oils and their derivates can be used as EP additives for lubricants. To conduct the investigations, sulfurized additives were synthesized using different vegetable-based oils and fatty acid esters, and alpha-olefins. In this study, the Four-ball test results were compared to gain a more accurate comprehension of how various raw-material-based additives influence wear and friction. The goal was to select raw materials that could be used with favorable results for the production of EP additives. The objective was to achieve a minimum Four-ball weld load parameter of 2000 N. The experiments revealed that the functional impacts of the synthesized samples are dependent on the type of raw materials employed. Based on the experimental data and the stated criteria, the examined raw materials were found to be suitable for the synthesis of EP additives.

## 1. Introduction

Modern lubricants use different additives to improve their different properties or to provide them with new, favourable properties, thus fulfilling the quality requirements for the lubricant [1]. There are many types of additives; in addition to EP additives, there are detergent-dispersant additives, corrosion inhibitors, oxidation inhibitors, and others [2,3]. The purpose of anti-wear and EP additives is to reduce wear and prevent seizures even at very high temperatures and pressures [4,5,6].

The value and field of application of vegetable oils depend on their fatty acid composition; since each vegetable oil has different physical and chemical properties, does its field of application. For example, oils containing C12 side chains (e.g., lauric acid) are important raw materials for the production of detergents and surfactants, while oils containing C18-22 side chains (e.g., oleic acid) are used in lubricants or as polymer additives [7,8].

In terms of regions, the Asian market dominates both production and consumption. Rapeseed oil is the most widely used vegetable oil as a bio-lubricant on the European market, while sunflower oil and soybean oil have greater importance in the US. In addition, the roles of castor oil, corn oil and safflower oil are small [8].

The linear structure gives the lubricant an anti-corrosion, anti-foaming, anti-wear and anti-friction effect. Due to these properties, they are excellent raw materials for high-temperature and high-pressure lubricants such as hydraulic fluids and EP additives. Wax esters are neutral lipids that are solid at room temperature and have limited availability in nature (cachalot oil, jojoba oil) [8].

Wax esters are oxo-esters of long-chain fatty acids and long-chain fatty alcohols. Wax esters of natural origin are mixtures of esters but also contain hydrocarbons. Wax esters have excellent performance properties due to their high oxidation stability and good resistance to hydrolysis. For this reason, stable additive compounds can be synthesized even with the use of EP additives [9,10].

The use of vegetable oils as possible substitutes or additives for different petroleum industrial applications makes the knowledge of the characteristics of these biomasses and their behavior of great importance at present. Some of the latest research focuses on ways to produce biomass-derived, multifunctional additives. [9,11,12,13].

In the current phase of the research, we aimed to examine the range of vegetable oils that can be used for the production of sulfurized vegetable oil additives based on the literature data and to compare them based on properties that have key importance for the synthesis of EP additives. Furthermore, the goal was to select vegetable oils that can be used with favorable results for the production of EP additives based on sulfurized vegetable oils. The targeted Four-ball weld load parameter had a minimum of 2000 N and the aimed weld scar diameter was set to a maximum of 0.50 mm.

## 2. Materials and Methods

### 2.1. Materials

Raw materials containing at least one double-bond in their molecular structure are suitable for the synthesis of sulfur-containing additives. Accordingly, raw materials that met this criterion were selected for the experiments. Vegetable oils, fatty acids, fatty acid methyl esters and alpha-olefins were selected. Throughout the experiments, examinations were conducted to determine how the unique compositions of these raw materials impacted the efficiency of the additives synthesized with their use.

Table 1 contains the data of the raw materials (vegetable oil and vegetable oil derivatives, and olefins) used for the synthesis of EP additives.

Vegetable oils are built of triglycerides, which are esters of various fatty acids with glycerol. This diverse structure affects the physicochemical parameters and functional effect of the additives synthesized with their use. Table 2 contains the data on the carbon atom number and double bonds of fatty acids typically found in vegetable oil triglycerides [14,15,16,17,18].

Based on the area of application and purity, the examined raw materials were found to be suitable for further investigations.

### 2.2. Synthesis

For the synthesis of the additive samples, the vegetable-origin raw materials or the alpha olefins were charged into the reactor in the first step. The sulfur powder was loaded into the continuously stirred unit, and the mixture was heated to a temperature higher than the melting point of orthorhombic sulfur. After a homogeneous mixture was formed, the reaction mixture was heated to the reaction temperature (120–190 °C). The reaction temperature was chosen depending on whether the formation of intramolecular (120–160 °C) or intermolecular (160–190 °C) reactions was preferred [3].

The detection and description of the reactions taking place are extremely complex due to the unique characteristics of the system. The first reaction step in this process is the S_8_-ring opening of the elemental sulfur and the subsequent oxidative attack of the sulfur on the vinylic protons (Equation (1)). This uncontrolled reaction ends up releasing H_2_S and leads to the formation of various sulfur-containing compounds. Most of the in situ-generated H_2_S is directly adsorbed by the double bonds, thus producing saturated alkyl-mercaptans (Equation (2)). The end-product comprises a comprehensive assortment of organic sulfur derivatives (Equation (3)). Among them, certain derivatives remain unsaturated, featuring isomerized double bonds and conjugated, chromophoric sulfur compounds like thioketones and thiophenes. These compounds cause the dark black colour of the product and its distinctive odour [3].
R – HC = CH – CH_2_ – R + S_x_ —› R – HC = CH – CH(SR) – R + HS_(x−1)_ + H_2_S(1)
R – HC = CH – R + H_2_S —› R – H_2_C – CH(SH) – R(2)
2R – H_2_C – CH(SH) – R + S_x_ —› R – H_2_C – CHR – S – (S_x−1_) – S – CHR- CH_2_ – R + H_2_S(3)

### 2.3. Methods

The detailed physical and physicochemical investigation of the examined raw materials used the following methods:Appearance [Visual].Density at 20 °C, [ISO 12185:1998], g/cm^3^.Kinematic viscosity at 40 °C [ISO 3104:1996], mm^2^/s.Kinematic viscosity at 100 °C-on [ISO 3104:1996], mm^2^/s.Dynamic viscosity at 40 °C-on [calculated] [ISO 3104:1996], mPa·s.Acid value [ISO 660:2000], mg KOH/g.Saponification number [ISO 6293:1994], mg KOH/g.Iodine value [ISO 3961:2000], g I_2_/100g.Water content [ISO 12937:2001], wt%.Sediment content [IEC 60422], wt%.

The so-called Four-ball method was applied to measure the functional effect of synthesized additives, which is widely used in the lubricant industry. The purpose of this testing is to determine the load-carrying capabilities of lubricating greases and oils under high-load applications, like bearings.

The Four-ball test machine functions by employing a rolling motion. This involves a single stainless steel ball rotating against three stainless balls, all coated with a lubricant film. The three stationary balls are held in place using a cradle. The load is incrementally increased until the lubricant film is depleted, resulting in metal-to-metal contact. Finally, the load is further increased until welding occurs. Based on the measured weld load result, lubricants can be formulated with varying levels of extreme-pressure properties. The following method was used to evaluate the extreme-pressure properties of lubricants with the Four-ball machine [19]:Four-ball test – weld load [DIN 51350-4:2015], N.

The Four-ball test machine was also utilized to assess the wear scar properties and coefficient of friction of a lubricant. This test aims to determine the ability of a lubricant to prevent wear. In the test, a steel ball is rotated against three fixed and lubricated steel balls under specified conditions of load, speed, temperature, and duration. The effectiveness of the lubricant in preventing wear is indicated by the size of the wear scar on the three stationary balls. After the test, the three wear scars are measured, and the average value is reported. The following method was used to evaluate the anti-wear properties of lubricants with the Four-ball machine [20]:Four-ball test – wear scar diameter [DIN 51350-5:2015], mm.

The main characteristics of the weld load and wear scar diameter measurements are summarized in Table 3.

The Four-ball tester equipment used for the examinations is shown in Figure 1. The schematic illustration of the principle of the measurements is also shown in Figure 1.

## 3. Results

The investigations were started by determining the physical and chemical properties of raw materials. After that, lubrication technical properties were determined. Recommendations were made for the application of the examined vegetable oils and their derivatives for EP synthesis.

The physical and physicochemical test results for the main qualification criteria are given in Table 4.

According to the measured results, a kinematic viscosity value that is more than twice as good as a significantly higher acid number value, between 3 and 3.6 mg KOH/g, which can be beneficial in terms of synthesis, was determined for the used cooking oil (UCO). However, it should be noted that the quality of the used cooking oil can fluctuate, so it is not recommended to use this vegetable oil derivative by itself for the synthesis of EP additives.

The examined olefins with different carbon chain lengths can be said to have a high iodine value, which is beneficial from the perspective of synthesis. As the carbon chain length increases, the iodine value decreases if the number of double bonds is unchanged.

The additive samples were made with the reaction of the raw materials characterized above and elemental sulphur. The sulfurization reaction took place in the liquid phase at about 140–170 °C and about 1–2 bar. To increase the reaction yield, mixing for 2–6 h is necessary at the reaction temperature [3,9,21,22,23].

To measure the functional effect of the synthesized additives, lubricating oil samples were made with the use of the Group I type base oil with 3 wt% additive content. Table 5 contains the results of the Four-ball weld load and scar diameter measurements.

The application of used cooking oil did not result in an EP additive with beneficial properties. On the other hand, the results of the additive sample synthesized using the upper phase obtained after separation of the higher-density fraction are more favourable. In many cases, the use of used cooking oil can be beneficial [13].

It was found that additives made from palm oil, olive pomace oil, grapeseed oil and C16C18 α-olefin have the best scarring characteristics. Differences in the wear scar diameter values were seen, but in none of the examined cases was the wear scar diameter greater than 0.58 mm.

Therefore, the presented vegetable oils and their derivatives, produced by the same technology, were investigated and resulted in effective EP additives (2100–2600 N).

However, there is a noticeable distinction in the outcomes; it is advisable to explore and apply a method through additional research that further differentiates the samples based on their functional effect.

Based on the data in Table 5 and Figure 2, we found that it is not possible to select vegetable oil and its derivatives based only on chemical composition.

Considering the results, the functional effect of the additives with the most favourable wear scar diameter results is shown in the figure below (Figure 2).

The best EP effect results were obtained with the use of palm oil, olive pomace oil and C12C14 α-olefin.

Considering the economics and availability conditions in addition to the results, palm oil, olive pomace oil, rapeseed oil and used cooking oil are recommended for further investigation after the development of the synthesizing technology (application of catalyst, optimization of technological parameters).

Based on the experimental data, the examined raw materials were found to be suitable for the synthesis of EP additives, because all the synthesized samples meet the targeted weld load criteria.

## 4. Conclusions

Almost all the raw materials used for synthesizing collected sulfurized vegetable oil type (EP) additives that can be found in the EU are accessible to different extents. The presented qualification properties can be considered decisive from the perspective of the synthesis. The relationships between individual properties were examined. The raw materials that were analyzed proved to be appropriate for producing EP additives since all the synthesized samples met the specified weld load requirements.

The results and conclusions can be used for the selection of raw materials that will be used for additive synthesis with elemental sulphur and dark sulphurization technology. In future research work, the use of other sulfur donors and application test methods will be investigated.

Based on a set of comparative experimental studies, the following conclusions can be drawn.

All basic physicochemical characteristics of the examined materials evaluated in this work present some relation with the properties of the tested vegetable oils and their derivates and olefins (where there is a relation between the viscosity and iodine value of materials).It was found that vegetable oil with a density above 0.913 g/cm^3^ is preferable for the synthesis of EP additives.The application of the upper phase of the used cooking oil is beneficial after sedimentation of its higher-density fraction.Although it is not possible to select vegetable oil and its derivatives based on their chemical composition only, it is preferable to use raw materials with a high iodine value, which is beneficial from the perspective of synthesis.It was found that additives made from palm oil, olive pomace oil, grapeseed oil and C16C18 α-olefin have the best wear scar-inhibiting effect. Therefore, the sulfurized derivatives of the investigated raw materials produced by the same technology resulted in effective EP additives (2100–2600 N).Based on the experimental data and the stated criteria, the examined raw materials were found to be suitable for the synthesis of EP additives. The most favourable outcomes in terms of the EP effect were achieved when utilizing palm oil, olive pomace oil, and C12C14 α-olefin. However, in the case of use, if a low wear scar diameter value is an important requirement for the additive, it is recommended to use the raw materials listed in Conclusion point 5.

## Figures and Tables

**Figure 1 materials-16-06570-f001:**
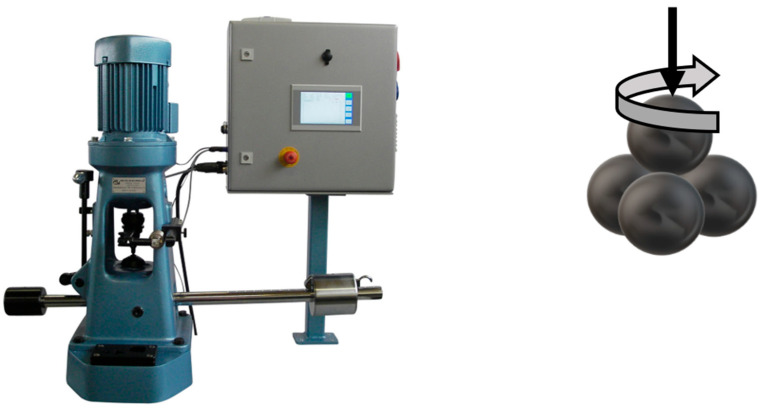
Four-ball tester and the principle of measurements.

**Figure 2 materials-16-06570-f002:**
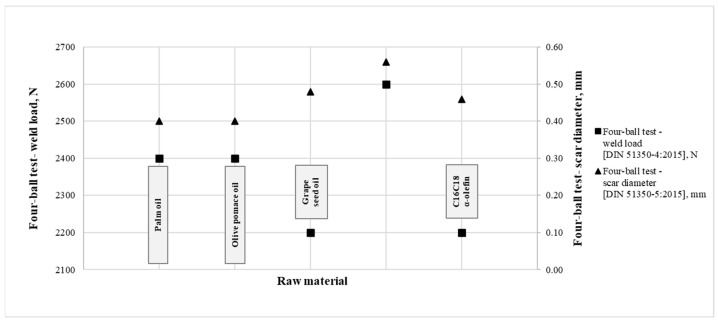
Four-ball weld load and Four-ball scar diameter parameters of the synthesized additives.

**Table 1 materials-16-06570-t001:** Properties of the used raw materials.

Properties	Rapeseed Oil (Food) 6404/2110	Sunflower Oil (Industrial) 6405/2110	Sunflower Oil (Food) 6331/2206	UCO ^1^ 6371/2109	UCO (Upper Phase) ^2^ 6194/2203	Palm Oil 6202/2203	Olive Pomace Oil 6133/2202	Extra Virgin Olive Oil 6538/2209	Grape Seed Oil 6539/2209	Castor Oil 6552/2209	Oleic Acid 210625/033	Stearic Acid 18:65 6197/2203	Fatty Acid Methyl Ester 6406/2110	C12C14 α-Olefin 6326/2205	C14C16 α-Olefin 6325/2205	C16C18 α-Olefin 6324/2206
Trade name	Refined rapeseed oil	Raw sunflower oil	Refined sunflower oil	UCO	UCO	Palm frying oil	Olive pomace oil	Bio Organic extra virgin olive oil	Refined grapeseed oil	Castor oil	Palmera A1818	Cremer AC Stearic acid 18:65	FAME B-100	Alpha Olefin C1214	Alpha Olefin C1416	Alpha Olefin C1618
Producer or distributor	Bunge Ltd., Budapest, Hungary	Bunge Ltd., Budapest, Hungary	Bunge Ltd., Budapest, Hungary	Biofilter Ltd., Törökbálint, Hungary	Biofilter Ltd., Törökbálint, Hungary	Vog Ltd., Bük, Hungary	Fortis Holding Ltd., Budapest, Hungary	Borges S.A.U., Tàrrega, Spain	Virgin Oil Press Ltd., Budapest, Hungary	Thermo Fisher Scientific, Waltham, USA	Palm-Oleo Sdn. Bhd., Petaling Jaya, Malaysia	CREMER OLEO GmbH, Hamburg, Germany	Rossi Biofuel Ltd., Komárom, Hungary	INEOS Oligomers, Alvin, TX, USA	INEOS Oligomers, Alvin, TX, USA	INEOS Oligomers, Alvin, TX, USA
Area of application	food industry, fuels	lubricants, lubricant additives, fuels	food industry, fuels	biodiesel production	-	food industry, chemical synthesis	food industry	food industry, cosmetics	food industry, cosmetics	medications, cosmetics, lubricants, chemical synthesis	lubricants, cosmetics, plastics	detergents, cosmetics, lubricants, coating	mainly biodiesel	detergent intermediates	oilfield and paper chemicals	oilfield and paper chemicals
Purity, %	100	100	100	100	100	100	100	100	100	100	min. 70	min. 65	min. 99.7	min. 99.7	min. 99.5	min. 99.2

^1^: Cooking oil that was used (mixture of rapeseed oil, sunflower oil, palm oil, and animal fat). ^2^:The upper phase of the used cooking oil (mixture of rapeseed oil, sunflower oil, and palm oil).

**Table 2 materials-16-06570-t002:** Fatty acid compounds of the investigated raw materials.

Raw Material	Carbon Atom Numbers and Double Bonds of Fatty Acids								
	C16:0	C16:1	C18:0	C18:1	C18:2	C18:3	C20:0	C20:1	Others ^3^
Rapeseed oil	4–5	0.3	1–2	56–64	20–26	8–10	0.6	1.4	0–1
Sunflower oil	4.2–6.8	0.3	2.1–5	20–25	63–68	0.2	0.4–1	-	0.3
Used cooking oil	19–21	4–5	4–6	50–55	0–1.5	0–1	0–1	0–1.5	0–1.5
Palm oil	37–44	0.4	3–6	39–44	8–10	0.3	0.4	-	1
Olive oil	7.5–20	0.3–3.5	0.5–5	55–83	3.5–21	0.9	0.4	0.4	0–1
Grape seed oil	5–7	0–1	3–5	13–15	73–77	0–1	0–0.5	-	2–4
Castor oil	0.5–1	-	0.5–1	4–5	2–4	0.5–1	-	-	83–85
Oleic acid	5–6		1.2–1.8	78–82	12–13	0.1–0.3	-		0–1
Stearic acid	31–34	-	62–68	0–0.5	-	-	0–0.5	-	0–2
Fatty acid methyl ester	8–9	0.5	2–4	51–60	21–24	6–8	0.5	1.2	1–2

^3^: Components with a carbon number less than C16:0 and/or greater than C20:1.

**Table 3 materials-16-06570-t003:** Properties of the two Four-ball test methods.

Measurement	Four-Ball Weld Load	Four-Ball Wear Scar Diameter
Equipment	Four-ball tester	Four-ball tester
Method	DIN 51350-4	DIN 51350-5
Size, shape (static)	d = 12.7 mm, ball	d = 12.7 mm, ball
Rotational frequency, shape (dynamic)	1500 rpm, ball	1500 rpm, ball
Load	max. 12,000 N	400 N
Measurement length	60 s	1 h
Measured property	weld load, N	wear scar diameter, mm
Contact type of surfaces	ball-ball	ball-ball

**Table 4 materials-16-06570-t004:** Physical and chemical properties of vegetable oils, vegetable oil derivates and olefins.

Method	Rapeseed Oil (Food) 6404/2110	Sunflower Oil (Industrial) 6405/2110	Sunflower Oil (Food) 6331/2206	UCO ^4^ 6371/2109	UCO (Upper Phase) ^5^ 6194/2203	Palm Oil 6202/2203	Olive Pomace Oil 6133/2202	Extra Virgin Olive Oil 6538/2209	Grape Seed Oil 6539/2209	Castor Oil 6552/2209	Oleic Acid 210625/033	Stearic Acid 18:65 6197/2203	Fatty Acid Methyl Ester 6406/2110	C12C14 α-Olefin 6326/2205	C14C16 α-Olefin 6325/2205	C16C18 α-Olefin 6324/2206
Appearance [Visual]	yellow, clear liquid	yellow, slightly opalescent liquid	yellow, clear liquid	brown, opalescent liquid	brown, clear liquid	light yellow, solid	yellow, clear liquid	yellow, clear liquid	light yellow, clear liquid	light yellow, clear liquid	dark yellow, clear liquid	white, solid	brownish yellow, clear liquid	colourless, clear liquid	colourless, clear liquid	colourless, clear liquid
Density at 20 °C [ISO 12185:1998], g/cm^3^	0.9166	0.9195	0.9187	0.9198	0.9202	0.9185	0.9141	0.9163	0.9183	0.9578	0.8955	not measurable	0.8797	0.7634	0.7789	0.7862
Kinematic viscosity at 40 °C [ISO 3104:1996], mm^2^/s	35.44	33.21	32.28	73.96	41.40	72.01	40.42	38.28	31.01	250.30	19.77	not measurable	4.36	1.44	2.28	2.95
Kinematic viscosity at 100 °C [ISO 3104:1996], mm^2^/s	8.11	7.81	7.74	13.94	8.84	8.76	8.45	8.39	7.60	19.19	4.85	not measurable	1.98	0.74	1.04	1.26
Dynamic viscosity at 40 °C [calculated] [ISO 3104:1996], mPa·s	32.04	30.13	29.26	67.13	37.59	65.27	36.45	35.08	28.48	239.74	17.45	not measurable	3.78	1.08	1.74	2.27
Acid value [ISO 660:2000], mg KOH/g	0.12	1.58	0.07	2.92	3.51	0.19	0.19	0.21	0.28	0.54	200	208.23	0.44	<0.01	<0.01	<0.01
Saponification number [ISO 6293:1994], mg KOH/g	191.9	191.1	191.2	194.0	194.2	198.8	189.1	198.4	204.1	178.2	199.1	209.68	193.8	0.0	0.0	0.0
Iodine value [ISO 3961:2000], gI_2_/100g	108.0	122.0	125.5	97.2	85.4	52.2	72.0	78.0	134.7	81.1	100.6	0.16	95.9	145.7	122.6	110.8
Water content [KF-Coulometry] [ISO 12937:2001], wt%	0.044	0.057	0.029	0.400	0.130	<0.01	0.050	0.070	0.010	0.190	0.08	not measurable	0.064	0.010	0.010	0.010
Sediment content [IEC 60422], wt%	<0.01	<0.01	<0.01	<0.01	<0.01	<0.01	<0.01	<0.01-								

^4^: used cooking oil (mixture of rapeseed oil, sunflower oil, palm oil, and animal fat). ^5^: the upper phase of used cooking oil (mixture of rapeseed oil, sunflower oil, and palm oil).

**Table 5 materials-16-06570-t005:** Active sulfur content and lubrication technical parameters of additives derived from the vegetable-origin raw materials and olefins.

Raw Material	Active Sulfur Content [ASTM D 1662], wt%	Four-Ball Test- Weld Load [DIN 51350-4], N	Four-Ball Test- Scar Diameter [DIN 51350], mm
Rapeseed oil, food grade	1.37	2300	0.50
Sunflower oil, industrial grade	1.18	2300	0.51
Sunflower oil, food grade	1.23	2300	0.51
Used cooking oil	1.94	2100	0.52
Used cooking oil (upper phase)	2.61	2300	0.50
Palm oil	4.70	2400	0.40
Olive pomace oil	3.78	2400	0.40
Extra-virgin olive oil	4.79	2300	0.50
Grape seed oil	1.62	2200	0.48
Castor oil	inhomogeneous solution		
Oleic acid	6.96	2300	0.58
Vegetable stearic acid 18:65	inhomogeneous solution		
Fatty acid methyl ester	4.72	2100	0.56
C12C14 α-olefin	9.37	2600	0.56
C14C16 α-olefin	7.89	2300	0.52
C16C18 α-olefin	6.45	2200	0.46

## Data Availability

No new data were created.

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
