# Peer review of "Investigation of Vegetable Oils and Their Derivatives for the Synthesis of Extreme Pressure Additives"

_materials, 2023, doi:10.3390/ma16196570_

Round 1

Reviewer 1 Report (Previous Reviewer 4)

1. I suggest authors to check Table 6, and correct measurement unit for movement speed!

2. Tables 7, 8, 9 and 10 might be better combined into one, thus improving visibility and clarity of the results.

3. Why is it important to give the views in figures 2, 3 and 4, if they do not provide any important information for research. The data on the absence of any correlation could be extracted from the unified table showing the physical and chemical properties.

4. Were there any limitations in the research work?

5. The conclusion does not state what are the further directions or possibilities of research.

Author Response

Thank you so much for your helpful feedback and suggestions. We carefully considered all of them and have made changes to enhance and expand upon the manuscript.

You will find the revised manuscript attached, along with our responses to your comments.

Reviewer 2 Report (Previous Reviewer 1)

Abstract:

Rewrite Abstract paragraph

- The abstract have to include : 1.Background, 2.Aim,3. Methods, 4.Results and Discussion and 5. Conclusions

- What is the percentage of the effect of olive and palm oil compared to other oils in this paper? Have to It is mentioned in the abstract

-Keywords:

Rewrite keywords

2.Materials and Methods

2.1 Materials

-It is prefer to write a brief summary of the oils materials (raw materials) which used in the paper and then show the tables (1-4).

-Table 1. Data  properties of vegetable oils

-Tables (1-3) cannot have the same name

- It is prefer to collect of tables (1-4) into one table,  because they are the same contents,  in landscape orientation paper size

-Line 65:

(vegetable oil, vegetable oil and their derivatives , and olefins)

-Do not use pronouns such as (our) ...... line 59-153

Results

-Tables 7-9 cannot have the same name

-It is prefer to collect of tables (7-10) into one table, because they are the same contents,  in landscape orientation paper size

-In Figure (5), the test results for palm oil and olive oil (Four-ball weld load and Four-ball scar diameter) appeared to be equal in values? Despite the difference in chemical composition (have to mention the scientific explanation)

-Testing of the material C16C18 α-olefin in Figure 5 was mentioned twice and showed different results for the same material (C16C18 α-olefin) Why?

- In figure (5) density, g/cm3 g/cm3

- In figure (4) kinematic viscosity at 40 C , mm2/s mm2/s

Conclusion

-No. 3: The name of the vegetable oil (palm oil) must be mentioned.

-No. 4 cancel it

-Re-conclusions based on the results achieved

Notes

- The appropriate tribological application of palm oil and olive oil must be mentioned, and a real lubricating application should be carried out in mechanical parts to determine their ability to withstand pressure and temperatures.

- Palm oil and olive oil must be compared with commercially applied industrial oils, to show us which one has the best  in tribological characteristics.

-This paper have to include the theoretical part which is represented by theoretical mathematical engineering equations related to this paper.

Author Response

Thank you so much for your helpful feedback and suggestions. We carefully considered all of them and have made changes to enhance and expand upon the manuscript.

You will find the revised manuscript attached, along with our responses to your comments.

This manuscript is a resubmission of an earlier submission. The following is a list of the peer review reports and author responses from that submission.

Round 1

Reviewer 1 Report

1- The researcher needs to compare palm oil and olive oil (the oils selected by the researcher) with commercial industrial oils used in machines and cars to find out what these oils are equivalent in terms of specifications (similarity)

Or what are the properties similarities between commercial industrial machine oils and palm and olive oils, in order to be able to use them instead.

2- Palm and olive oils need real testing inside the engines to find out their resistance to Friction, Wear and Fatigue at pressure and temperatures.

3- The research needs the theoretical part within the investigation or simulation using simulation programs.

Author Response

Thank you very much for your critique and valuable suggestions.

Related to the comments I would like to feedback that testing the lubricants made by the synthesized additives in industrial applications is not part of the research at the current phase, in agreement with the industrial partner.

In this research, the described raw materials are used to synthesize sulfurized additives and they are not used by themselves instead of commercially industrial base oils or lubricants.

Reviewer 2 Report

I could not find any technical correction required for this manuscript to be published in the journal submitted.  A fine spell check of the writeup details needed.

Author Response

Thank you very much for your kind review.

Reviewer 3 Report

Manuscript No.: Materials, 1607489

Date received February 16, 2023

Title: Investigation of vegetable oils and their derivatives for the 2 synthesis of Extreme Pressure additives

Authors: Gábor Zoltán NAGY, Roland NAGY

According to the Abstract the paper investigated the rapeseed oil and soybean oil the vegetable which oils can be used as the raw material, improving  their different properties or to give them a new, favorable property, thus fulfilling the quality requirements for the lubricant of EP additives.

After carefully reviewing this paper, I recommend that it:

In this subchapter, the authors talk about “The so-called four-ball method was applied to measure the functional effect of synthesized additives, which is widely used in the lubricant industry.” it is necessary to present the tribometer and the method, not just to mention it.

The work is very general, it takes into account many oil samples, but the investigations are simple and at the level of a laboratory work, not a scientific research work at all.

I consider that the work is not valuable enough to be published as a scientific work in this Materials journal.

None of methods can be considered original.

For all the above the presented work in this form needs improvement and the decision is to reject the paper.

Author Response

Thank you very much for your critique and valuable suggestions. Based on these, the article has been modified as follows.

The manuscript has been supplemented with a description and a summarizing table about the applied methods. Pictures of the tribometer and the measurement principle have been inserted.

Analysing raw materials with precision holds significant importance, driven by advancements in technology and industrial processes. In the current phase of the research, we found it advisable to employ international methods which are used and continuously developed in the industry. Furthermore, the examined samples were derived from certain uncommon raw materials, which yielded promising outcomes.

Reviewer 4 Report

The names of pictures from 1 to 3 must be formulated in such a way that they precisely describe what is shown in the picture.

In line 159, a sentence is given in which it is not clear why the good result of Grape seed oil on the Four-ball scar test was omitted.

Nowhere in the paper is it explained why the authors concluded that oils with a density greater than 0.913 g/cm3 are preferred for the synthesis of EP additives (conclusion no. 2).

Conclusion number 6 is not clearly formulated. I think it should be more specific in the context of stating exactly which oils are suitable for the synthesis of EP additives, based on experimental results.

Author Response

Thank you very much for your critique and valuable suggestions. Based on these, the article has been modified as follows.

The names of the pictures have been supplemented to describe more precisely what is shown in the pictures.

The conclusion has been updated with the results of the samples derived from grapeseed oil and used cooking oil.

Our conclusion regarding density was drawn based on the functional effect assessment results.

Conclusion number 6 has been supplemented and reformulated. In addition, the values of the expected functional effects were included among the objectives of the research.

Reviewer 5 Report

This work must be absolutely rejected because the authors have shown an unforgivable negligence by writing an Abstract that is identical to the one of another article on the same argument recently published by the same authors on another Journal: Acta Materialia Transylvanica 5/2(2022)83-88. How it is possible to write the same Abstract for two different papers? Moreover, some other parts of the text submitted are identical to the one of the mentioned article. This is not acceptable for the reputation of  a scientific journal like Processes. However, the scientific content of this article is very poor. A long list of commercial properties of different vegetable oils is reported in Tables 1-4. These Tables seem not useful for the objective contained in the title of the work. In Table 5 are reported the chemical properties  of 10 different oils of common use. These properties are largely known and reported in many papers and books. In the paragraph entitled "Methods" is reported a list of routinary physical methods for characterizing oils. These methods have probably used for collecting the data reported in Table 6,7,8,9. The conclusions of the authors about all these reported data is that "no relashionship can be found between the examined properties". Therefore Figures 1,2 and 3 are not useful results. The only acceptable results seem to be the ones reported in Table 10. In this table the Four Balls Tests for different sulfurized oils are reported. As it can be seen, from this Table the difference between all the examined systems is relatively small. Figure 4 is just a graphical repetition of some of the data of Table 10. No data are reported about the degree of sulfurization and no chemical investigation has been made about the chemical structure of sulfurized system and properties. For all the above reported reasons my suggestion is to reject this work.

Author Response

Thank you very much for your critique and valuable suggestions. Based on these, the article has been modified as follows.

The abstract has been kindly revised to further align with the content of the revised manuscript.

The subject of the cited Acta Materialia Transilvanica article is about the examination of sulphurizable raw materials, based on information available in the literature. In the experiments presented in the current manuscript, the raw materials utilized for the synthesis were carefully chosen, taking into account the results of the research work carried out in connection with the referenced previous article. Due to the shared topic and interdependence of the two articles, there are identical references in the introductions. However, their source has been indicated, and they rightfully hold a place in both articles.

The parameters specified for the raw materials were derived from thorough analytical tests conducted following international industry standard methods (with the exception of the typical fatty acid compositions presented in Table 5). As a result, we conscientiously acquired precise knowledge about the properties of the raw materials employed, rather than solely relying on widely available general results from the literature. These findings offer substantial advantages in establishing the operational parameters for syntheses and represent a fundamental requirement for producing effective additive compounds.

However there is a noticeable distinction in the outcomes, we agree that it is advisable to explore and apply a method through additional research that further differentiates the samples based on their functional effect. We have already started the research on this topic.

The figure about the Four-ball weld load and Four-ball scar diameter parameters of the synthesized additives (Figure 5) contains the four best results only for the purpose of emphasis. The table about active sulfur content and lubrication technical parameters of additives derived from the vegetable origin raw materials and olefins (Table 11) additionally contains all of the results.

Data about the active sulfur content of the synthesized additives, basic chemical information about this type of additives and some information about the process parameters of the technology used for laboratory production are inserted.